# New Route to the Production of Almond Beverages Using Hydrodynamic Cavitation

**DOI:** 10.3390/foods12050935

**Published:** 2023-02-22

**Authors:** Cecilia Faraloni, Lorenzo Albanese, Graziella Chini Zittelli, Francesco Meneguzzo, Luca Tagliavento, Federica Zabini

**Affiliations:** 1Istituto per la Bioeconomia, CNR, Via Madonna del Piano 10, 50019 Sesto Fiorentino, Italy; 2HyRes S.r.l., Via Salvator Rosa 18, 82100 Benevento, Italy

**Keywords:** almond, almond beverage, almond skin, antiradical activity, green extraction, hydrodynamic cavitation, nutritional values, polyphenols, proteins, vitamins

## Abstract

Perceived as a healthy food, almond beverages are gaining ever-increasing consumer preference across nonalcoholic vegetable beverages, ranking in first place among oilseed-based drinks. However, costly raw material; time and energy consuming pre- and posttreatments such as soaking, blanching and peeling; and thermal sterilization hinder their sustainability, affordability and spread. Hydrodynamic cavitation processes were applied, for the first time, as a single-unit operation with straightforward scalability, to the extraction in water of almond skinless kernels in the form of flour and fine grains, and of whole almond seeds in the form of coarse grains, up to high concentrations. The nutritional profile of the extracts matched that of a high-end commercial product, as well as showing nearly complete extraction of the raw materials. The availability of bioactive micronutrients and the microbiological stability exceeded the commercial product. The concentrated extract of whole almond seeds showed comparatively higher antiradical activity, likely due to the properties of the almond kernel skin. Hydrodynamic cavitation-based processing might represent a convenient route to the production of conventional as well as integral and potentially healthier almond beverages, avoiding multiple technological steps, while affording fast production cycles and consuming less than 50 Wh of electricity per liter before bottling.

## 1. Introduction

The consumption of plant-based beverages has rapidly grown in recent years, partially replacing dairy products in the diet for a variety of reasons including health (lactose intolerance, cholesterol, and blood glucose level issues), lifestyle choices, or ethical and environmental concerns. A 10.4% increase in worldwide sales of plant-based beverages is expected from 2018 to 2023, reaching USD 26 billion per year [1]. Plant-based beverages are aqueous extracts of cereals, legumes, nuts, seeds, and pseudo-cereals [2], showing a wide variety of nutritional properties and micronutrients. Almond-based beverages, along with rice-based beverages, contributed most to the 380% volume increase in the consumption of rice/grain/nut/seed-based beverages in Europe from 2012 to 2015 [3]. Additionally, almond (*Prunus dulcis* L.) has been the most produced nut worldwide in recent years [4], gaining economic significance in the global food supply chain.

Soy and almond beverages have received special consideration due to their good nutritional profile and potential biological functions [5]. Almond beverages showed interesting compositional characteristics in terms of monounsaturated fatty acids content and balanced composition in the content of proteins, fat, fibers, and vitamins [6], although the specific abundance of macro- and micronutrients, in the absence of any additives other than water, depends on the composition of raw materials, i.e., skinless almond kernels, which show a large variability across varieties, climates, growing practices, and harvesting season [7]. Almond beverages can have a remarkable content in vitamin E, a fat-soluble antioxidant that can protect cells from the harmful effects of free radicals towards cancer and cardiovascular diseases [8]. Almonds are indeed important sources of mono- and unsaturated fatty acids, minerals, vitamin E, polyphenols, and phytosterols, with antioxidant properties that can have beneficial effects on human health [9]. Robust evidence exists about the association of almond consumption with various health benefits [10], including improvements to the metabolic system [11,12], microbiota [13], and cardiovascular system [14,15], as well as antioxidant, anti-inflammatory, anticancer, antimicrobial [4], and antidiabetic activity [16].

While kernels, representing around 52% of the total fresh weight, are by far the most used component of almond for human consumption, other parts (skin, shell, hull, etc.) are often discarded, despite their interesting properties and their disposal representing an important environmental burden [4]. Almond skin, representing around 4% of the total weight of the almond, was shown to possess beneficial properties. Phytochemicals and polyphenols contained in almond skin were associated with antibacterial and antiviral effects [17,18] to the scavenging of free radicals, and were proven to induce quinone reductase [19].

Although the specific processing steps allowing the manufacturing of plant-based beverages depend on the physiology of the particular vegetable matrix, they aim invariably at the maximum possible yield of soluble extract. For this purpose, almond seeds, sometimes after roasting, require peeling as a basic step, since skin removal allows an efficient release of kernel’s nutrients and micronutrients into water, despite the loss of important skin micronutrients [20].

For the purpose of peeling, further industrial steps are required such as soaking in water and hot water blanching, followed by wet milling, homogenization, and pasteurization or sterilization [20]. To the best knowledge of the authors, no substantial innovation has been applied in recent times to this production process, whose steps are described in greater detail in Section 2, where new technologies, aimed at replacing thermal treatments on resulting almond extracts, are also introduced.

This study presents the first evidence of the possibility of adopting controlled hydrodynamic cavitation (HC), an emerging green, efficient, and scalable method for the extraction of natural products [21], as a single-unit operation, thus replacing all the other traditional production steps in the extraction in water only of almond kernels, including whole seeds (seeds including the skin), to produce beverages at concentrations matching the market standards, as well as more concentrated extracts ready for further dilution. Performance data, including extraction yields and comparison with a high-end market product, process time, and specific energy consumption, are provided in order to allow both replication and comparison with other methods.

## 2. Technological Overview

As mentioned in Section 1, traditional industrial production steps of almond beverages are consolidated and did not substantially change in the last few decades, including [1,20,22,23]

Roasting (optional, to increase the emulsion stability and the solubility of proteins): 95–100 °C, 30 min;Soaking in water: 4 °C, 6 h;Blanching with peeling: in water, 90 °C, 3 min; in steam bath, 85 °C, 5–30 min.Wet milling: 1:9 almond to water mass ratio, 18,000 rpm, 2 min;Filtration from solid residuals;Possible addition of stabilizers such as gums, sweeteners, salt, hydrocolloids, emulsifier, or fortified with micronutrients such calcium or some vitamins;Homogenization and sterilization (deactivation or extermination of spoilage or pathogenic microorganisms): ultrahigh temperature (UHT), 140 °C, few seconds; ultrahigh-pressure homogenization (UHPH), 350 MPa, 85 °C (with many variants).

A few of the above-listed processing steps, such as roasting, hot water blanching and skin removal (peeling), and sterilization/homogenization by means of UHT or UHPH, are particularly energy intensive, and may negatively affect rheological and nutritional characteristics of products, the latter, for example, through the partial denaturation of almond proteins and the change of the profile of fatty acids, and might be harmful to valuable micronutrients, such as polyphenols of both almond skin and skinless kernel [24,25,26].

The peeling step deserves a special mention, also due to the high value of almond skin. As an alternative method working at room or moderate temperatures, ultrasound-assisted extraction (UAE) was tested and validated on the laboratory scale [25], but not yet at the preindustrial scale. UAE, whose effectivity is largely based on cavitation phenomena induced in the irradiated medium, was also applied successfully to enhance the extraction rate of nutrients and micronutrients from few vegetable materials and the stability of the aqueous extracts, including almonds [27,28]. However, intrinsic limitations of UAE, also due to the rapid attenuation of ultrasound waves in liquid media, make its full scalability hard to achieve, in fact leaving HC as the only feasible option for large-scale applications, across methods based on cavitation processes [29]. With similar outcomes to UAE [30], HC shows key advantages, including easy scale-up, lower capital cost, and higher efficiency, such as in the case of peanut milk production [31].

More in general, the evolution of consumption toward greater attention to the healthy properties of food and environmental sustainability stimulated an accelerated search for more effective and efficient technological solutions in the food supply chain. Process time, energy consumption, reduction of food waste, and preservation of healthy components have been the steering factors of this search since at least mid-1990s’, including plant-based beverages and, in particular, oilseed beverages, which have long been perceived since as potentially healthy products and are usually composed of more than 90% water [2].

New technologies have been developed and tested, aimed at overcoming the shortfalls affecting conventional production processes, while ensuring enhanced chemical-physical stability and microbiological safety. Table 1 lists and shortly describes the most relevant emerging technologies with special focus on almond beverages.

To the best knowledge of the authors, not only none of the above studies, but the relevant emerging technologies in general, have ever been applied to the extraction of almond seeds, but only to finished almond beverages manufactured according to conventional methods. Moreover, in all the studies cited in Table 1, as well as in any other studies, almond seeds were peeled before manufacturing the almond beverages.

Hydrodynamic cavitation (HC) technologies and related methods are emerging among the most effective, efficient, and straightforwardly scalable in the field of the extraction of natural products, not only in comparison to newest green technologies but also to conventional methods [21]. The most important properties of HC-based extraction methods derive from the unique capability of concentrating the energy of mixed liquid-solid fluxes into microscopic hot spots with extremely high energy density, in turn released at the collapse of cavitation bubbles in the form of mechanical and thermal energy, as well as from the relatively straightforward design and set-up [37]. Thus, HC methods have been proposed as important tools to help achieving the sustainability development goals in few different technical fields [38]. HC methods have shown high process yields as single-unit operation systems applied to the extraction of natural products in water only at the preindustrial scale [39], such as in the brewing field (extraction of cereals and hops), involving starch, proteins, and polyphenols as the main constituents released into the water phase [40], conifer tree parts, involving polyphenols and volatiles [41], waste citrus peel involving pectin, polyphenols and volatiles [42], and soybean, involving proteins and fat [43].

The application of HC methods to the manufacturing of almond beverages, with perspectives up to the industrial scale, was already devised by Meneguzzo et al. in 2020 [21]. The rest of this study aims at providing the first proof at the pilot scale of such a new route to the production of almond beverages.

## 3. Materials and Methods

### 3.1. Production of Aqueous Almond Extracts

Almond (*Prunus Dulcis*) skinless kernels in the form of flour (<1 mm in size) and fine grain (1–2 mm in size), both from the same batch of almond seeds, and whole seeds (including the skin) in the form of coarse grain (about 3–5 mm in size) were supplied by the company Dolceamaro S.r.l. (Monteroduni, IS, Italy). All materials came from the variety Lauranne^®^Avijor grown in Italy, which is a recently released late-flowering cultivar [44], with kernels characterized at least since 2010 [45]. Figure 1 shows a picture of the whole almond seeds in the form of coarse grain.

The almond materials were extracted in tap water only, with different concentrations. The details of the batch hydrodynamic cavitation-based extractor, comprising a closed hydraulic circuit of total volume around 200 L, with a centrifugal pump and a Venturi-shaped reactor with circular section as the key components, and electricity as the only energy source, were described in a previous study about the extraction of waste orange peel [42]. Pump’s impellers transferred mechanical energy to the liquid–solid mixture, in turn converting into heat during the process, and no heat dissipation method was used. Absorbed power and energy consumption, in the form of electricity supplied to the centrifugal pump, were measured by means of a three-phase digital power meter (IME, Milan, Italy, model D4-Pd, power resolution 1 W, energy resolution 10 Wh, accuracy according to the norm EN/IEC 62053-21, class 1).

Table 2 shows for each extraction test the test ID, the type, mass and concentration (% of total weight) of almond material, the overall process time, and the temperature range of the process. Each almond batch was preserved at room temperature for no more than 2 days after reception. Almond materials were inserted into the extraction system all together at the beginning of each process (initial temperature levels 26–34 °C).

Tests MFP1 and MGP1 were performed with a concentration of 7.4%, which is comparable with a high-end product available on the market, “Valdibella al naturale” (Valdibella agricultural cooperative, Camporeale, PA, Italy), hereinafter also referred to as the “commercial product”. The commercial product had a concentration of 8% and, at the time of the tests MFP1 and MGP1, had been packaged in Tetra Pak^®^—Tetra Brik^®^ Aseptic about 1 month before and showed an expiration date after about 9 months. The commercial product was manufactured with organic almonds from the typical Sicilian (Italy) varieties Tuono, Genco, Supernova, Pizzuta, and Fascionello, few of which were characterized with regards to the phenolic content, fatty acids, proteins, and volatiles [4]. The commercial product was manufactured based only on almonds cream from skinless kernels and water, according to a patented procedure (text in Italian only) [46]. The analytical figures derived for the commercial product were normalized according to the ratio of the concentration used in tests MFP1 and MGP1 to that of the commercial product, i.e., each content was multiplied by the factor 7.4/8.0 = 0.925. Finally, it is noteworthy that the comparison among the samples collected from the experimental tests and the commercial product was aimed at a preliminary check of the compliance of the experimental products with a high-end market standard, while too many uncertainties remain about the commercial product, including the almond varieties actually used, the respective harvesting season, and the details of the manufacturing process, preventing further investigation.

Tests MGP2 (from the same batch of almond seeds as test MGP1) and MGP3 were performed at higher concentration (27.2% and 18%, respectively), aimed at showing both the possibility of obtaining concentrated extracts than could be subsequently diluted to create beverages suitable for the market, and, with test MGP3, to verify the possibility of extracting whole coarsely ground almond seeds, the latter in turn with the double purpose of avoiding the peeling step before the extraction and exploiting the bioactive properties of almond skins.

### 3.2. Sampling and Microbiological, Nutritional, Total Polyphenols, and Antiradical Activity Analyses

Table 3 shows the types of analyses performed for each extraction test and the high-end commercial almond beverage. Storage issues with samples collected from the test MGP2, except those collected for the assessment of polyphenols and antiradical activity, prevented the respective microbiological and nutritional analyses.

#### 3.2.1. Sampling

For test MFP1, the aqueous extracts were sampled at the temperatures of 40 °C, 47 °C, 58 °C, 68 °C, and 74 °C. For test MGP1, the aqueous extracts were sampled at the temperatures of 40 °C, 47 °C, 58 °C, 68 °C, 78 °C, and 86 °C. After filtering with a 200 µm sieve (stainless steel mesh), three samples were collected at each point in sterile bottles, each 500 mL in volume. The sterile bottles were stored at −20 °C until analysis. For tests MGP2 and MGP3, the aqueous extracts were sampled only at the end of the process, at the temperature of 82 °C. The same volumes of the commercial beverage were collected from the respective packaging at the beginning of the analyses. For test MGP3, three further samples, each one in a Falcon test tube of volume 50 mL, were collected at the end of the process without filtration (integral extract), aimed at the assessment of the mass balance. Moreover, 0.5 kg of all raw materials used in the extraction tests were preserved for further analyses.

#### 3.2.2. Microbiological, Nutritional, and Vitamin Analyses

Microbiological, nutritional, and vitamin analyses were performed by laboratories accredited by Accredia, The Italian Accreditation Body (https://www.accredia.it/en/, accessed on 6 January 2023; laboratory No. 0069 L and laboratory No. 0792l), complying with standards UNI CEI EN ISO/IEC 17025 and ISO 9001.

The colony count of total microorganisms at 30 °C was measured according to UNI EN 12822:2000 (https://store.uni.com/en/uni-en-iso-4833-1-2013, accessed on 6 January 2023); the concentrations of yeasts and molds were measured according to UNI EN 21527-1:2008 (https://store.uni.com/en/iso-21527-1-2008, accessed on 6 January 2023).

Each nutritional quantity was analyzed according to a specific method:Energy level: EU Regulation No. 1169/2011 of the European Parliament and of the Council of 25 October 2011 on the provision of food information to consumers (https://eur-lex.europa.eu/legal-content/EN/ALL/?uri=CELEX%3A32011R1169, accessed on 2 January 2023);Total and unsaturated fat: Istisan report No. 1996/34 “Methods of analysis for the chemical control of foods”, pages 39 and 47, respectively (https://www.iss.it/en/rapporti-istisan, accessed on 6 January 2023);Total carbohydrates and sugar: Italian Ministerial Decree 03 February 1989 (https://www.gazzettaufficiale.it/eli/id/1989/07/20/089A3049/sg, text in Italian, accessed on 6 January 2023);Protein: Istisan report No. 1996/34 “Methods of analysis for the chemical control of foods”, page 17 (https://www.iss.it/en/rapporti-istisan, accessed on 6 January 2023);Fiber: Istisan report No. 1996/34 “Methods of analysis for the chemical control of foods”, page 73 (https://www.iss.it/en/rapporti-istisan, accessed on 6 January 2023);Vitamin B2 and vitamin PP: AOAC 2015.14-2015 (http://www.aoacofficialmethod.org/index.php?main_page=product_info&cPath=1&products_id=2990, accessed on 6 January 2023);Vitamin E: UNI EN 12822:2000 (https://store.uni.com/en/uni-en-12822-2000, accessed on 6 January 2023).

#### 3.2.3. Total Polyphenols and Antiradical Activity

The total phenolic content (TPC) was determined based on the Folin–Ciocalteau method [47], modified according to the AOAC SMPR 2015.009 (https://www.aoac.org/resources/smpr-2015-009/, accessed on 6 January 2023), using gallic acid (Sigma-Aldrich) as standard. The analyses were performed in triplicate. However, it was anticipated that the recovery and quantification of polyphenols from raw almond materials during laboratory analysis is critically dependent on the details of the method and could be affected by greater systematic uncertainties than declared [24], an issue that does not affect the analysis of clear aqueous extracts. As further pointed out in Section 4, such concern was confirmed by the results obtained in this study, involving also vitamins; thus, the measurements of the content of polyphenols and vitamins performed on raw almond materials were discarded from the analysis.

The antiradical activity was performed according to [48]. DPPH (2,2-diphenyl-1-picrylhydrazyl) (Sigma-Aldrich, St. Louis, MO, USA) is a stable radical that can be reduced by reaction with an antiradical hydrogen–donor compound. A spectrophotometer (Beckman DU-640, Fullerton, CA, USA) was used to measure this colorimetric reaction at 517 nm, when the color of the DPPH radical changes from violet to yellow. The methanolic extracts were diluted at different proportion to find the concentration at which 50% of initial absorbance value of sample with added DPPH is obtained. In order to determine the absorbance at 517 nm, 1 mL of diluted extract was added to 1 mL of methanol DPPH solution (63 M), mixed, and measured right away. After 20 min, the absorbance was tested again. A decrease of 50% in the initial DPPH concentration is referred to as IC50, which is the concentration inhibiting 50 % of DPPH radicals.

For each extract, the IC50 was calculated with the following formula: % inhibition = [100 − (Ax/As)] × 100,(1)
where As is the initial absorbance of the extract sample in DPPH solution (t = 0) and Ax is the absorbance of the same sample after 20 min. At least 4 different concentrations of the extracts were used to determinate the IC50. The analyses were performed in triplicates.

#### 3.2.4. Potential Contents

Based on nutritional contents measured for raw materials, rough estimates of the respective potential contents in the aqueous extracts were computed, calculating the total amount of each relevant quantity (concentration multiplied by mass of the raw material) and dividing the result by the volume of water used in the extraction tests. Potential contents were used for a preliminary assessment of the extraction yield of each considered relevant quantity.

### 3.3. Mass Balance

The integral extract (without filtration), collected at the end of test MGP3 in three Falcon test tubes, each of volume 50 mL, was preserved at –20 °C until analysis, then thawed at room temperature and centrifuged at 9980 g for 1 h at 15 °C (ALC multispeed refrigerate centrifuge mod PK131R), to obtain a solid fraction (pellet), and an aqueous fraction containing smaller particles (supernatant). Centrifugation time longer than 1 h did not increase the sedimentation rate of small particles, which remained as suspension in the supernatant. Pellet and supernatant were dried in an oven (Type M40-VN, MPM Instruments S.r.l., Bernareggio, Italy) at 70 °C until constant weight (almost 4 days). The analysis was performed in triplicate.

## 4. Results

### 4.1. Processes and Energy Consumption

Figure 2 shows the processes in terms of time, temperature, and specific energy consumption (assessed as the energy consumption per liter of extracted beverage), including the indication of sampling points.

Based on the comparison of tests MGP1, MGP2, and MGP3, it is notable that the specific energy consumption is not practically sensitive to the raw material concentration; thus, since the dilution step would not add relevant energy consumption, the production of concentrated extracts in operational environments would be effective for energy saving.

### 4.2. Microbiological Stability

Figure 3 summarizes the results of the microbiological analyses at time zero and after preservation for 7 days at the temperature of 4 °C in sterile bottles (shelf life), limited to the colony count of microorganisms at 30 °C (hereinafter also referred to as “microorganisms”). The concentrations of yeasts and molds (not shown) were very low, and in any case below the limit of detection of 9 UFC/g in the samples collected at process temperature of 68 °C or higher, both at time zero and at shelf life, while in the commercial product the concentration of yeasts was below the limit of detection at time zero and 27 (7–110) UFC/g at shelf life. The results for tests MFP1 and MGP1 are compared with the commercial product (red lines).

Almond kernel flour used in MFP1 and fine grain used in MGP1 had counts of microorganisms at the level of 10^4^ (range 2.2 × 10^3^ to 4.6 × 10^4^ CFU/g) and 2.5 × 10^4^ CFU/g (range 5.6 × 10^3^ to 1.1 × 10^5^ CFU/g), respectively, while counts of microorganisms for coarse grain of whole almonds, used in test MGP3, was at the level of just 3.5 × 10^2^ CFU/g (range 2.5 × 10^2^ to 4.9 × 10^2^ CFU/g).

For samples collected from tests MFP1 and MGP1, the counts of microorganisms were never significantly higher than that of the commercial products, both at time zero and at shelf life, when the count of microorganisms for the commercial product increased by about 1.8 logs compared to time zero. For the same samples, the count of microorganisms decreased with the process temperature at the sampling point, in particular monotonically, with temperature for test MFP1 at shelf life, reaching a level significantly lower than the commercial products for the sample collected at 74 °C, almost identical to the level measured at time zero (on average 1800 CFU/g). For test MGP1, a remarkable decrease in the count of microorganisms occurred at the sampling temperature of 78 °C, which is especially visible at the shelf life, when count levels fell below that of the commercial product (about 4 × 10^4^ CFU/g compared with about 2 × 10^5^ CFU/g, although with large uncertainties).

For test MGP3, the count of microorganisms at time zero in the sample collected at 82 °C was just 41 CFU/g (range 30 to 56 CFU/g), reflecting the very low contamination level in the raw material.

### 4.3. Nutritional Contents

#### 4.3.1. Tests MFP1 and MGP1

Figure 4 summarizes the results of the nutritional analyses at time zero for test MFP1, compared with the potential contents (green lines) and the contents measured for the commercial product (red lines).

Figure 5 summarizes the results of the nutritional analyses at time zero for test MGP1, compared with the potential contents (green lines) and the contents measured for the commercial product (red lines).

Energy contents increased with time and temperatures for both tests MFP1 and MGP1; however, there was a more definite trend for test MGP1, where they eventually reached the potential content and the energy content of the commercial product, which were practically indistinguishable. Since fat represents by far the most abundant macronutrient of almonds, it is not surprising that the fat contents followed the same trend as the energy contents, reaching the potential contents and the energy content of the commercial product at the end of the processes. Contents of the saturated fat were very low, matching the potential content and the content of the commercial product at the end of the processes.

The content of carbohydrates in the samples from both tests MFP1 and MGP1 remained approximately constant across time and temperature, on average close to 80% of both the content observed for the commercial product and the potential content. The content of sugars is not shown due to very high uncertainties; however, the only sugar above the level of detection (0.10%) was sucrose, which remained practically constant in all the samples (in test MGP1, starting from the sample collected at the temperature of 47 °C).

The protein content for test MFP1 remained practically constant starting from the first collected sample, indistinguishable from the content observed for the commercial product and approximately 75% of the potential content. The protein content for test MGP1 increased slightly over time, eventually reaching at the end of the process on average 80% and 65% of the content of the commercial product (from which it was indistinguishable) and the potential content, respectively.

The content of fibers in the samples from both tests MFP1 and MGP1 remained almost constant, in particular starting from the samples collected at the temperature of 58 °C, at much lower levels (on average between 18% and 25%) than in the commercial product, however strictly matching the respective potential contents.

#### 4.3.2. Test MGP3

Table 4 summarizes the results of the nutritional analyses at time zero for test MGP3, performed on the sample collected at the end of the process at the temperature of 82 °C. The potential contents are also shown.

Extraction yields for all the nutritional quantities were substantially lower, on average half, than for test MGP1 at comparable temperatures, with the exception of proteins (extraction yield of 58% against 65%). The extraction yield for fibers was particularly low, about 20%.

### 4.4. Total Polyphenol Content and Antiradical Activity

#### 4.4.1. TPC in Tests MFP1 and MGP1

Figure 6 summarizes the results of the TPC analyses for tests MFP1 and MGP1, which are compared with the commercial product (red lines).

In the samples collected during test MFP1, TPC was about 55% of the content found in the commercial product up to the sample collected at the temperature of 58 °C, then increased to 60% at 68 °C and 74 °C.

In test MGP1, TPC increased to 73% of the content observed in the commercial product in the sample collected at the temperature of 68 °C, then exceeded the commercial product on average by 1.5 times in the last sample collected at the temperature of 86 °C, although with a large uncertainty.

TPC was also measured at shelf life, i.e., after preservation for 7 days at the temperature of 4 °C (data not shown). No significant changes occurred, either in the commercial product or in the test samples, except a reduction of TPC in the sample collected from test MGP1 at the temperature of 86 °C, which decreased on average by 23% from the initial content, however remaining about 8% higher than in the commercial product, from which it was practically indistinguishable.

#### 4.4.2. TPC and Antiradical Activity in Tests MGP2 and MGP3

Table 5 summarizes the results of the TPC analyses for the samples collected at the end of the tests MGP2 and MGP3, along with the respective IC50 levels for the antiradical activity of the aqueous extracts, assessed according to the DPPH essay.

The ratio of average TPC observed for test MGP2 to the same quantity observed for test MGP3 was 1.37, which is far lower than the ratio of the respective almond mass contents, i.e., 1.70 (the water volume being the same). Thus, the polyphenols released in the water phase in test MGP3 were higher than in test MGP2, on average by about 24% after normalizing to the respective almond mass contents.

The IC50 level showed even more interesting and surprising results. In fact, in test MGP3 it was almost 35% lower than in test MGP2, possibly suggesting that whole almond seeds used in test MGP3 provided not only more polyphenols but also more active ones with regards to the antiradical activity. A hypothesis that might explain this experimental evidence is advanced in Section 5.

### 4.5. Vitamins

Figure 7 summarizes the results of the analyses of vitamin B2 (Riboflavin), vitamin PP (Niacin + Niacinamide), and vitamin E, for samples collected from tests MFP1 and MGP1, which are compared with the commercial product (red lines).

The content of vitamin B2 in samples collected from both tests matched the content observed for the commercial product already from the first sample collected at 40 °C, remaining around the same content until the end of the process.

The content of vitamin PP for test MFP1 increased during the process up to about 60% of the content found in the commercial product for the sample collected at the temperature of 74 °C. For test MGP1, the content of vitamin PP increased up to a content indistinguishable from that of the commercial product at the temperatures of 78 °C and 86 °C.

The measurements of the content of vitamin E were affected by large uncertainties; thus, little can be said about the trends. However, the samples collected from both tests MFP1 and MGP1 starting at the temperatures of 74 °C and 78 °C, respectively, showed contents definitely above the detection level of 10 mg/kg and reaching about 14 mg/kg in test MGP1, thus suggesting the occurrence of its extraction in the water phase. In the commercial product, the content of vitamin E was below the level of detection.

### 4.6. Mass Balance for Test MGP3

Mass balance analysis was performed on the sample collected at the end of test MGP3 at the temperature of 82 °C. Table 6 summarizes the results.

As a proof of consistency, it can be noticed that the total dry biomass content (8.73 ± 0.42 g) accurately represents the content of the original raw material (concentration of 18%), as shown by the dry to fresh biomass ratio of 18.5%.

In an adequately clarified extract of whole almond seeds in the form of coarsely ground grain, i.e., the supernatant in the mass balance assessment shown in Table 6, more than 27% of the original dry biomass was transferred to the aqueous extract (2.38 g out of 8.73 g of dry biomass), while about 72% of the original dry biomass turned into dry pellet, i.e., the dry residue of the process (6.29 g out of 8.73 g). Such pellet had a moisture content of about 74% (6.29 g of dry biomass out of 24 g of fresh biomass), which could be reduced at the filtration/separation step in operational environments, thus increasing the mass yield of the clarified extract.

## 5. Discussion

This study provided the first evidence of the feasibility and potential advantages of a HC-based extraction system as a single-unit operation with industrial perspectives, for the production of almond beverages and concentrated aqueous extracts, both from skinless kernels and whole seeds.

Based on the results presented in Section 4.2, HC processes allowed achieving microbiological stability at much lower temperatures in comparison to conventional heat treatments, such as UHT, as shown also by means of the comparison with a commercial high-end product that undergone UHT treatment. A peak temperature level of 74 °C was found to be sufficient to ensure a total count of microorganisms in the shelf-life analysis below that observed for the commercial product, as well as the absence of molds and yeasts also at shelf life (preservation for 7 days at 4 °C). The microbiologic stability of the last sample collected at 74 °C from test MFP1 was surprising, showing a count of microorganisms at shelf life even lower than at time zero and suggesting that microorganism cells were no longer viable.

In principle, this achievement would allow for higher protection to thermolabile compounds in the manufacturing process of almond beverages, as well as for avoiding the sterilization step in the industrial production chain, with consequent savings in energy consumption. The use of whole almond seeds in test MGP3, with a count of microorganisms in the raw material much lower than in the other raw materials, allowed the production of an extract practically free of microorganisms. This was likely due to the long-known properties of the almond kernel skin, which represents a protective layer that prevents the oxidation and microbial contamination of the kernel [4].

Based on the results presented in Section 4.4.1, the substantial retention of TPC in the tests MFP1 and MGP1 at shelf life suggests the effective inactivation of the polyphenol oxidase enzymes during the process, as observed with HC-based treatment of blueberries [49] and sugarcane juice [50], thus contributing to the stability of the product. Rancidification can be an important issue for foods, such as almonds and derived products, which are rich also in polyunsaturated fatty acids [1]. While not measured objectively, no visual or olfactive sign of rancidification emerged at shelf life for the samples collected at peak temperatures during the tests, possibly suggesting an effective inactivation of the lipoxygenase enzymes, which was observed in the case of other emerging food treatment methods [23].

Overall, the results about the stability of HC-derived products represent an original achievement of this study, as well as the foundation for all the other results.

Based on the results presented in Section 4.3.1, the evolution of the energy content in the tests MFP1 and MGP1 closely followed the fat content, which is consistent because fat dominated over the other nutritional quantities. Potential fat contents and their contents in the commercial product were indistinguishable. Fat was extracted very fast in MFP1, so much that, after less than 20 min and at the temperature of 40 °C, its content was more than 70% the potential content, after which it barely changed until 68 °C, then increasing towards the potential content. The larger size of almond kernel grains used in test MGP1 likely delayed the extraction of fat, with a yield of about 55% after less than 20 min at 40 °C, after which it increased regularly up to 78 °C, then suddenly accelerated up to a fat concentration exactly matching the potential content.

The saturated fraction of fat followed approximately the same evolution as total fat, matching the potential content, along with the corresponding content of the commercial product (about 8.5% of the total fat content), at the same time and temperature. The retention of the original partition of the fat into the saturated and the unsaturated fractions is an important achievement, since unsaturated fatty acids, among other constituents of almond seeds, were attributed major health effects, such as decreasing blood lipid concentrations and neuroprotection [20].

It took about 2 h of process time and a peak temperature of 86 °C for the complete extraction of fat; although, based on the obtained results, higher initial temperatures, for example, following heat recovery at the end of the process, should not affect substantially the extraction of fat, provided that the peak temperature reaches at least the level of about 80 °C. Further research is recommended towards the optimization of the process.

The concentration of carbohydrates in the raw almond materials was about ten times lower than fat; thus, these macronutrients had comparatively lower relevance to the composition of the obtained extracts. However, the extraction of carbohydrates, along with the sugar fraction, was quite fast and practically indistinguishable from the potential level starting at the temperature of 58 °C.

The extraction rate of proteins in tests MFP1 and MGP1 closely resembled that of fat, showing very fast extraction with almond kernel flour and slightly slower with fine grains, however eventually converging around the level of the commercial product, at about 65% of the potential level. A hypothesis for such an incomplete extraction rate could be advanced, about a dynamical balance between the extraction and degradation rate of the proteins during the hybrid HC and thermal extraction processes. Such hypothesis is supported by previous research, which showed that partial heat-induced almond protein denaturation occurs already at temperatures between 45 °C and 55 °C and at an accelerated pace above 65–75 °C, while such proteins, although water soluble and thus in principle easily extractable by HC processes, are embedded in oleosins surrounding the oil droplets, making them harder to extract [51]. The matching of the obtained protein concentration levels with the commercial product also appears to support the above hypothesis, which might represent a general limitation in the production of almond beverages. However, further research on this topic is necessary, also following recent findings and recommendations [52].

Fibers were quickly extracted and, starting at the temperature of 58 °C, their concentrations strictly matched the potential levels for both tests MFP1 and MGP1; however, they were more than three times lower than in the commercial product, likely due to either a greater content of fibers in the raw material used to manufacture the commercial product, or the retention in that product of the entire content of almonds cream, as per the relevant patent [46].

Overall, with concentrations of almond skinless kernel material, in the form of flour (<1 mm in size) and fine grain (1–2 mm in size), typical of commercial almond beverages, the single-unit operation HC processing showed the ability to produce extracts that were microbiologically stable and preserved practically all the nutritional properties of the raw material, although some improvements could be tried for the extraction of proteins. Based on data shown in Section 4.1, the entire process from the mixture of water at room temperature and almond material to the output of the extract ready for filtration and packaging, lasted between about 100 and 120 min, with a specific energy consumption between about 60 Wh/L and 100 Wh/L.

Based on the results presented in Section 4.3.2, the extraction rates for test MGP3, using a concentrated mixture (18%) with whole almond seeds in the form of coarse grains (about 3–5 mm in size), were on average half of those achieved for the test MGP1 at comparable temperatures. The limiting factors might be either the coarser size, which took more time to HC processes for grinding and pulverization and left less time for extraction, or the protection offered by the skin to the extraction of substances embedded in the kernel, or the higher concentration of the almond raw material itself, which limited the frequency of interactions of solid particles with pressure shockwaves and mechanical jets generated at the collapse of the cavitation bubbles, or the combination of all the above factors. However, it is noticeable that the extraction rate of proteins in test MGP3 was only slightly lower than in test MGP1, which supports the above-presented hypothesis about the complex extraction/denaturation kinetics, i.e., late extraction of proteins in MGP3 might have limited their denaturation. Further experiments and theoretical research are recommended on this topic, for example to investigate the effects of isothermal steps on the extraction rate of nutritional compounds, especially at temperatures below the above-mentioned threshold for protein denaturation (45–55 °C) [51], or by using reactors able to generate more aggressive and effective cavitation regimes [37].

On the sensorial side, beyond the subjective judgment of the authors about the good taste of both the beverage-like extracts (tests MFP1 and MGP1) and the concentrated extract (test MGP3), the retention of the kernel skin in the latter test did not alter too much the usual creamy white color that consumers are used to, as shown in Figure 8.

Overall, the possibility of generating high concentration aqueous extracts from whole almond seeds, ready for further dilution and production of almond beverages, by means of HC-based processes as a single-unit operation, was successfully demonstrated, which is another original result of this study. The obtained extract was practically free of microbial contamination, although endowed with about half of the potential nutritional properties, thus requiring further research and process optimization.

On the sustainability side, based on Figure 2b, the consumption of specific energy of about 100 Wh/L at the end of the test MGP3 (concentration of 18%) would translate, after dilution, in a specific energy consumption for the almond beverage (concentration around 8%) of 50 Wh/L or even lower.

Finally, the existence of a critical gelation temperature, estimated at 87.5 °C, while representing an upper limit for the production of acceptable almond beverages or concentrated extracts ready for dilution, might offer the chance to generate new products by means of HC processes, such as almond tofu or cheese, which require higher concentrations of raw almond material than used for the manufacturing of commercial beverages [51].

The analyses performed on micronutrients extracted in the aqueous phase offer further elements to assess the performance of the HC-based processing system and the nutraceutical quality of the products.

Based on the results presented in Section 4.5, the HC-based extraction of vitamin B2 extremely was fast and effective, so much that its contents for both tests MFP1 and MGP1 matched the respective level in the commercial product already in the samples extracted at the temperature of 40 °C. The extraction rate of vitamin PP was quite fast, too, with contents close to the commercial product at 58 °C and matching it in test MGP1 at the temperatures of 78 °C and 86 °C. Such efficient HC-driven extraction was not surprising, due to the well-known high water-solubility of vitamins B2 and PP.

Contrary to the other considered vitamins, vitamin E is fat soluble and in principle harder to extract in water only. However, although delayed, its concentration in samples collected from tests MFP1 and MGP1 eventually exceeded the detection limit at the temperatures of 74 °C and 78 °C, respectively, while it was not detected in the commercial product. Based on the well-known heat sensitivity of vitamin E and its degradation beginning already at 40 °C [53], a hypothesis similar to the one presented for proteins can be advanced. In particular, a dynamical balance might occur between the extraction rate, hindered by the lipophilic nature of vitamin E, and its degradation rate, which might be partially corroborated by the late extraction and higher levels achieved in test MGP1, as well as by the slight decrease in its concentration in test MGP1 from 78 °C to 86 °C. Due to the relevance of vitamin E for human health, further experimental and theoretical research on this topic is recommended.

Based on the results presented in Section 4.4.1, TPC in tests MFP1 and MGP1 were practically indistinguishable already in the first sample collected at the temperature of 40 °C, and no further change occurred up to the temperature of 74 °C in test MFP1 and 68 °C in test MGP1, with contents in the range 50 to 60 mg/kg. The extraction rate accelerated in test MGP1 at 78 °C and even more at 86 °C, extracting about 60% of TPC in in the temperature range 68 to 86 °C and eventually exceeding the content observed in the commercial product. The bimodal structure of the extraction rate, with two peaks at the beginning of the processes up to 40 °C, and in the later phase (68 °C to 86 °C), is likely to reflect the complex composition of polyphenols of almond kernels, a few tenths of which were identified and characterized [9]. The extraction rate of those polyphenols from almond kernels, and thus their identification and quantification, was found to be remarkably dependent on the extraction method, such as the used solvent, temperature, etc. [9,24], hence the complex pattern of the extraction rate emerging from tests MFP1 and MGP1.

Based on the results presented in Section 4.4.2 and Table 5, not only was the average TPC in the samples collected at the end of the tests MGP3, after normalization of the almond seed mass, higher than in test MGP2, but the DPPH IC50 level of the MGP3 sample was far lower than for test MGP2, despite lower absolute TPC. It has long been known that almond skin, beyond representing a protective layer that prevents from the oxidation and microbial contamination of the kernel [4], is particularly rich in polyphenols and other bioactive compounds with remarkable antiradical activity [19,54], as well as antimicrobial and antiviral activities [4,17,18], which has prompted further studies aimed at exploiting the potential of almond skin as a byproduct of the peeling step of almond seeds [25]. These properties of the almond kernel skin are the most likely candidates to explain the substantial superiority of the antiradical activity of the aqueous extract from whole almond seeds.

Overall, the effective extraction of bioactive micronutrients by means of HC processes, both the main vitamins and polyphenols available in almond kernels was successfully demonstrated. Based also on previous and extensive evidence, especially concerning the HC-based extraction of polyphenols [55], this topic appears quite consolidated. However, the findings about the bimodal extraction rate of polyphenols (higher extraction rates early in the process at moderate temperatures and later at relatively high temperatures), the effective extraction of polyphenols from whole almond seeds, which was associated with a substantially higher level of antiradical activity, and the effective extraction of vitamin E, along with the hypothesis on the related mechanisms, represent further original results of this study. Future research might investigate, in particular, the topic of antiradical activity, which brings an important contribution to the healthy properties of the product, including more biologically relevant essays than the DPPH.

Finally, the mass balance information about test MGP3, provided in Section 4.6, could be economically relevant, both to derive the potential mass yield of clarified extracts (almond beverages or concentrated extracts), and because the pellet could be reused as a filler for food or feed products, possibly still endowed with a residual content of insoluble fibers and proteins, or conveyed to biodigesters for energy generation. Further research is suggested on the analysis of the pellet resulting as a byproduct from HC-based processing of almond raw materials.

Figure 9 shows a summary scheme of the developed method for the production of almond beverage, along with basic information about the quality of the produced beverage (macronutrients and micronutrients), its stability at shelf life, and energy consumption.

This study was affected by some important limitations, which are listed and shortly discussed below.

The design of experiments was not optimized due to limitations in the availability of raw material and other resources; information available from the performed tests was exploited as much as possible, and more structured experiments are planned. This study did not investigate the rheological properties of the aqueous almond extracts, which are fundamental to the physical stability (for example, sedimentation and phase separation) and acceptability of the products [1,6,51], while, based on previous research, it can be only hypothesized that HC processes help creating stable nanoemulsions, allowing to overcome physical stability issues even without any further additives [29].

This study neither investigated the effectivity of the extraction of amino acids, which have primary relevance to the nutritional quality of any food including almonds [10], nor the presence and activity of almond-derived allergenic compounds in the aqueous extracts, which can represent an important issue [56].

Further research is recommended on all the above-discussed topics, which were not investigated in this study.

## 6. Conclusions

Hydrodynamic cavitation-based processing of almond skinless kernels or whole seeds, as a single-unit operation, showed potential advantages over conventional techniques in the production of almond beverages or concentrated aqueous extracts. The same process was able to deal with different types of almond materials, i.e., skinless kernels in the form of flour or fine grain, and whole almond seeds, including the skin and in the form of coarse grain, to produce microbiologically safe and stable extracts, to effectively extract the most important macronutrients, although with margins of improvement in the case of whole seeds, and to very effectively extract all the most important bioactive micronutrients, with special advantages in the case of whole seeds, likely due to the bioactive properties of the almond kernel skin. The nutritional composition of the extracts was comparable with a high-end organic commercial product, while showing comparable or better microbiological stability and generally superior availability of bioactive micronutrients.

The proposed method, which is straightforwardly scalable to any production capacity, would allow replacing with a single operation the complex processing chain used in conventional manufacturing methods for almond beverages, including roasting (optional), soaking in water, blanching and peeling, wet milling, the addition of stabilizers (optional), homogenization, and sterilization. Additionally, the proposed method would allow for an advantageous processing of whole almond seeds, also reducing the environmental burden due to the removal and disposal of almond kernel skin. All the available figures, including process time, specific energy consumption, and mass balance, were provided, in order to allow for meaningful comparisons with other techniques, whether well-established or emerging ones.

## Figures and Tables

**Figure 1 foods-12-00935-f001:**
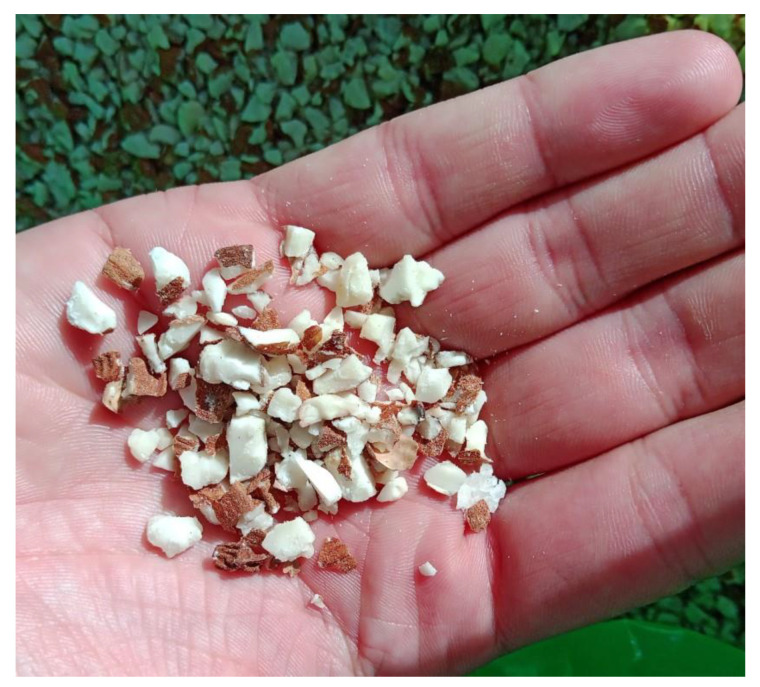
Sample of coarse grain of whole almonds.

**Figure 2 foods-12-00935-f002:**
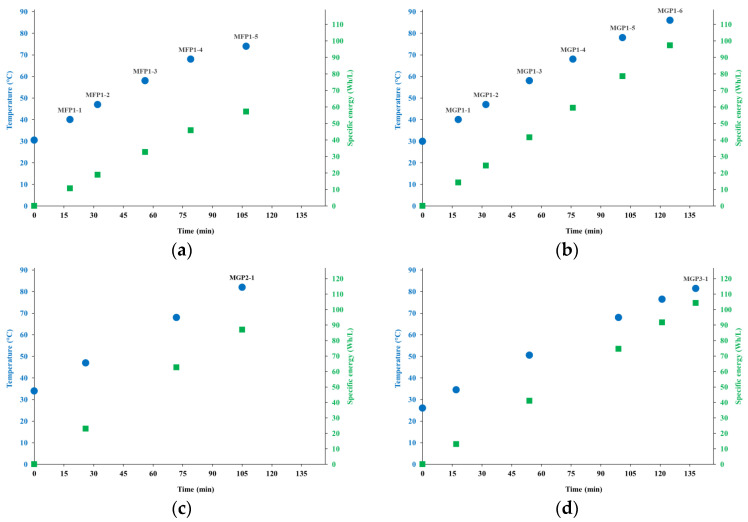
Process time, specific energy consumption and sampling points: (**a**) test MFP1; (**b**) test MGP1; (**c**) test MGP2; (**d**) test MGP3.

**Figure 3 foods-12-00935-f003:**
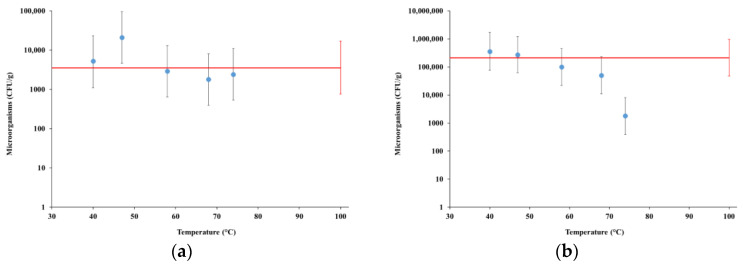
Concentration of microorganisms: (**a**) test MFP1 at time zero; (**b**) test MFP1 at shelf life; (**c**) test MGP1 at time zero; (**d**) test MGP1 at shelf life; (**e**) test MGP3 at time zero. CFU unit means colony-forming units, represented in logarithmic scale for better readability. Red lines represent the same quantity for the commercial product.

**Figure 4 foods-12-00935-f004:**
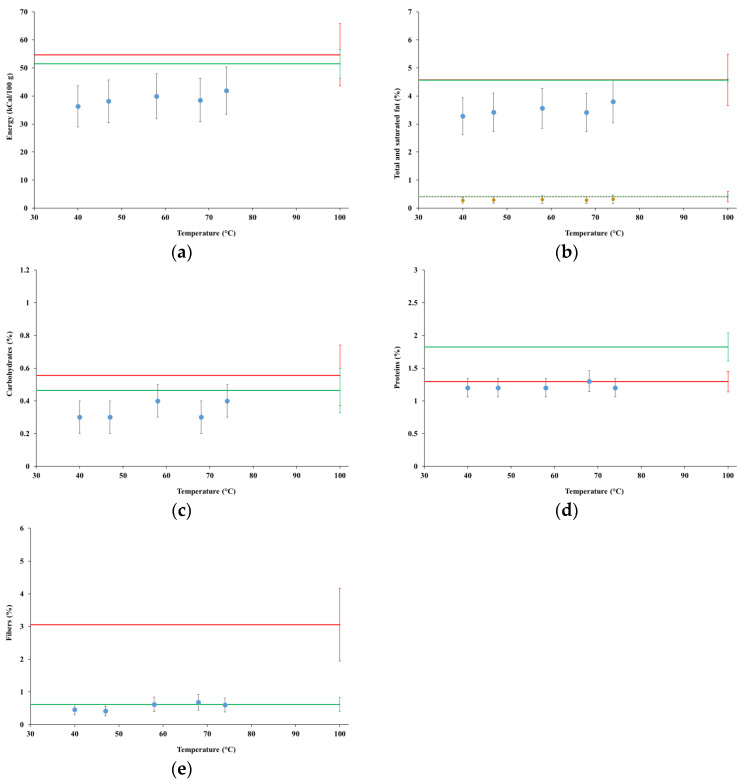
Nutritional contents for test MFP1: (**a**) energy; (**b**) total and unsaturated fat; (**c**) carbohydrates; (**d**) protein; (**e**) fiber. Red lines represent the same quantities for the commercial product; green lines represent the relevant potential contents. Dotted lines in (**b**) represent the content of unsaturated fat in commercial product (red) and the respective potential content (green).

**Figure 5 foods-12-00935-f005:**
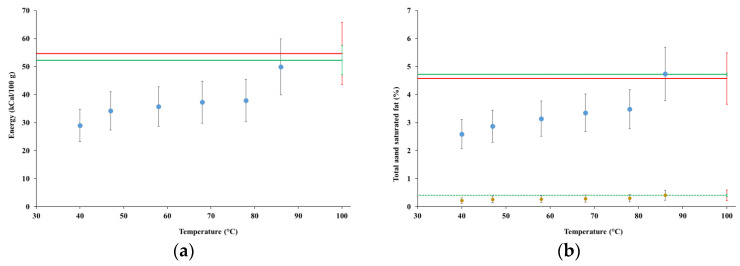
Nutritional contents for test MGP1: (**a**) energy; (**b**) total and unsaturated fat; (**c**) carbohydrates; (**d**) protein; (**e**) fiber. Red lines represent the same quantities for the commercial product; green lines represent the relevant potential contents. Dotted lines in (**b**) represent the content of unsaturated fat in commercial product (red) and the respective potential content (green).

**Figure 6 foods-12-00935-f006:**
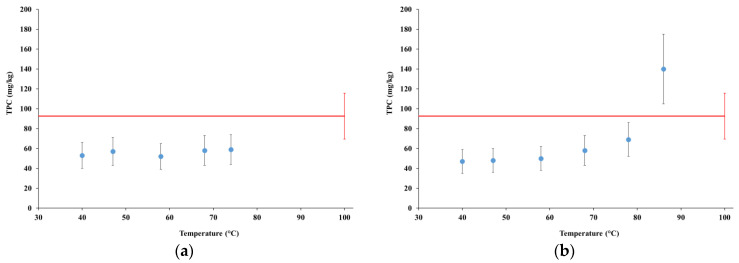
Total polyphenol content at time zero: (**a**) test MFP1; (**b**) test MGP1. Red lines represent the potential content.

**Figure 7 foods-12-00935-f007:**
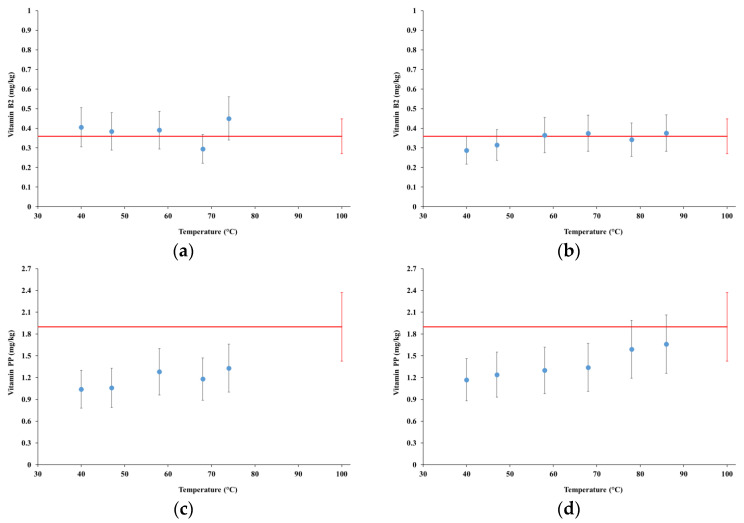
Concentration of vitamins B2, PP, and E in the extracts: (**a**) test MFP1, vitamin B2; (**b**) test MGP1, vitamin B2; (**c**) test MFP1, vitamin PP; (**d**) test MGP1, vitamin PP; (**e**) test MFP1, vitamin E; (**f**) test MGP1, vitamin E. Red lines represent the same quantities for the commercial product.

**Figure 8 foods-12-00935-f008:**
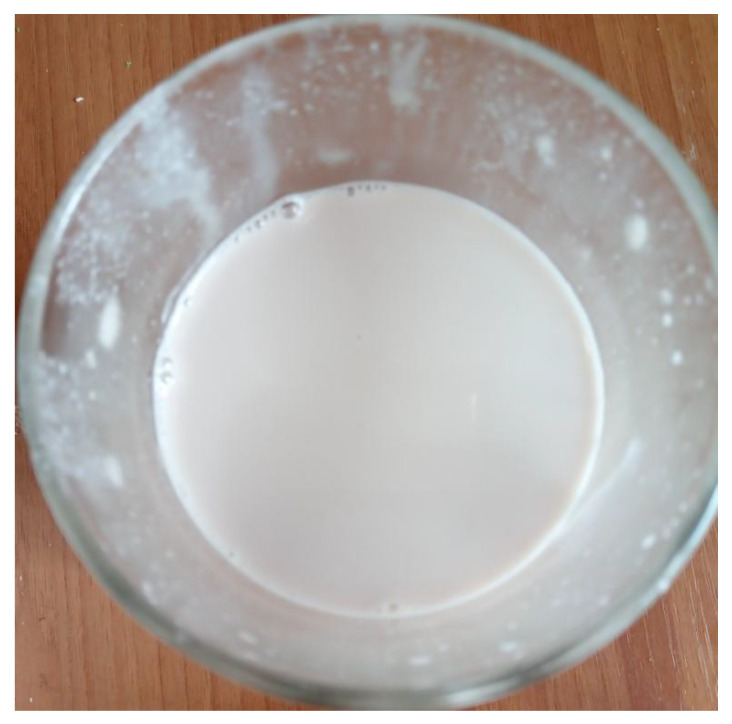
Visual appearance of the almond concentrated extract from test MGP3.

**Figure 9 foods-12-00935-f009:**
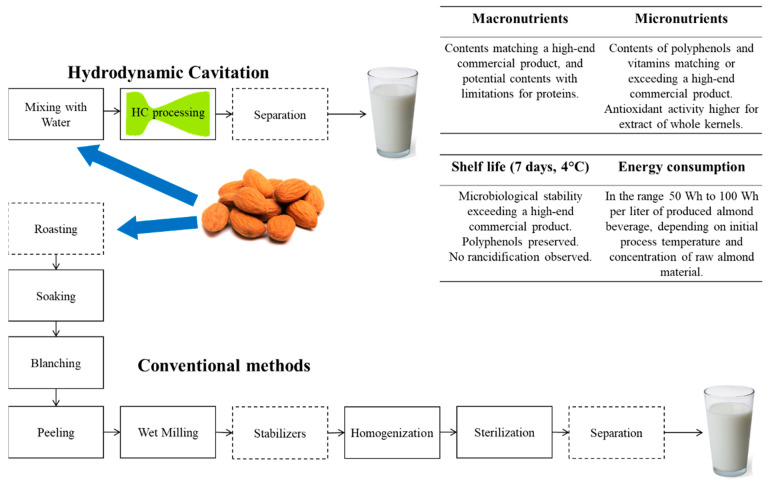
Summary scheme of the developed method for the production of almond beverage, along with comparative contents of macronutrients and micronutrients, shelf life stability properties, and energy consumption.

**Table 1 foods-12-00935-t001:** Emerging technologies for the treatment of almond beverages.

Technology	Application	Process	Main Results	Reference
US	Almond beverage: disinfection	130 W/80%/20 kHz 8 min/6 s of pulse	*Escherichia coli* (O157:H7): 5.12 to 3.81 log CFU/mL. *Listeria monocytogenes*: reduction by 1 log CFU/mL.	[32]
US	Almond beverage: physicochemical	300 W/20 kHz/100% 0 to 5 min	Higher Brix degree and physical stability. Decreased viscosity and suspended particles size.	[6]
HHP	Almond beverage: physicochemical	HHP (450 and 600 MPa for 0, 30, 60, 180, 300, and 600 s at 30 °C) Control: Traditional thermal process (0, 30, 180, and 300 s at 72, 85, and 99 °C).	Aggregation and coagulation of almond proteins. Improved sensorial properties.	[33]
HPP	Almond beverage: physicochemical	350 MPa and 85 °C for 15 s.	Microbiological stability. Increase of particle size. No change of cytotoxic, genotoxic, and antigenotoxic activity.	[34]
TS	Almond beverage: disinfection, physicochemical, micronutrients	TS: 600 W/40 kHz/30, 45, and 60 °C for 10, 20, 30, and 40 min Control: pasteurization (60 s at 90 °C)	Particle size reduction due to acoustic cavitation. Improvement of rheological properties. Increased bioavailability of phenolics.	[35]
UHPH	Almond beverage: disinfection, chemical–physical improvement	200 and 300 MPa at 55, 65, and 75 °C, with emulsifying agent (lecithin)	200 MPa with 55 °C inlet temperature improves over conventional pasteurization.	[5]
UHPH	Almond and soy beverages: physicochemical, microbiological, nutritional, and toxicological	200 MPa, 55 °C; 300 MPa, 75 °C. Control: UHT.	300 MPa, 75 °C led to a complete inactivation of microorganisms, improved colloidal stability	[36]

US = ultrasound; HHP = high hydrostatic pressure; TS = thermosonication; UHPH = ultrahigh-pressure homogenization.

**Table 2 foods-12-00935-t002:** Basic features of the extraction tests.

Test ID	Almond Material	Mass of Almonds (kg)	Water Volume (L)	Concentration (%)	Process Time (min)	Process Temperatures (°C)
MFP1	Peeled, flour	15	187.5	7.4	107	30–74
MGP1	Peeled, fine grain	12	150	7.4	125	30–86
MGP2	Peeled, fine grain	56	150	27.2	105	34–82
MGP3	Whole, coarse grain	33	150	18.0	138	26–82

**Table 3 foods-12-00935-t003:** Analyses performed on the experimental samples. A symbol “X” indicates a performed analysis.

Test ID	Sample Type ^1^	Microbiological ^2^		Nutritional	Total Polyphenols	Antiradical Activity
		T_0_	Shelf life			
MFP1	Raw	X		X	X	
MFP1	Extract	X	X	X	X	
MGP1	Raw	X		X	X	
MGP1	Extract	X	X	X	X	
MGP2	Extract				X	X
MGP3	Raw	X		X	X	
MGP3	Extract	X		X	X	X
Commercial	Extract	X	X	X	X	

^1^ Raw: raw material (solid); Extract: aqueous extract. One or more extracts were sampled from each test. ^2^ Some microbiological analyses were performed at time zero (T_0_) and after preservation of the sample for 7 days at the temperature of 4 °C (shelf life).

**Table 4 foods-12-00935-t004:** Nutritional contents in the sample from test MGP3 and potential contents. Where available, standard deviations are indicated.

Quantity	Sample MGP3	Potential Content
Energy (kCal/100 g)	52	119
Total fat (%)	4.3 ± 0.9	10.6 ± 2.1
Saturated fat (%)	0.3 ± 0.1	0.8 ± 0.2
Carbohydrates (%)	0.8 ± 0.2	1.8 ± 0.4
Proteins (%)	2.5 ± 0.5	4.3 ± 0.9
Fibers (%)	0.8 ± 0.2	3.9 ± 0.8

**Table 5 foods-12-00935-t005:** TPC and DPPH antiradical activity (IC50 level), in samples from tests MGP2 and MGP3.

Test ID	TPC in Extract (mg/kg)	DPPH IC50 (μL/mL)
MGP2	193 ± 20	42.93 ± 0.09
MGP3	141 ± 14	28.13 ± 0.73

**Table 6 foods-12-00935-t006:** Mass balance for test MGP3.

Material	Fresh Mass (g)	Average Ratio to Fresh Mass (%)	Dry Biomass (g)	Average Ratio to Dry Mass ^1^ (%)
Whole ^2^	47.3 ± 1.6	100.0	8.73 ± 0.42	18.5
Pellet	24.0 ± 0.9	50.8	6.29 ± 0.31	26.2
Supernatant	23.3 ± 0.8	49.2	2.38 ± 0.14	10.2

^1^ Ratio of dry biomass to original fresh mass of the relevant material. ^2^ Whole content of the Falcon test tube.

## Data Availability

The data presented in this study are available on request from the corresponding author.

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
