# Peer review of "New Route to the Production of Almond Beverages Using Hydrodynamic Cavitation"

_foods, 2023, doi:10.3390/foods12050935_

Round 1

Reviewer 1 Report

This study provided the evidence of the feasibility and potential advantages of a Hydrodynamic cavitation (HC)-based extraction system as a single-unit operation with industrial perspectives. However, some shortcomings exist in this study: the graphs were not clear enough, the analysis was not deep enough, the results were not clear enough and the article was not concise enough. It is recommended to add missing samples and relevant experiments for the missing samples, and to reconstruct the analytical ideas of the article. A major revision is recommended.

1.     Abstract:The description of the results should be more detailed.

2.     Introduction and discussion need to be concise and simplified.

3.     Materials: Missing samples need to be added (Table 2, MGP2) and related experiments.

4.     Results and discussion: The results are described and discussed in detail. The systematic errors caused in the experimental results should be further investigated and further interpreted.

5.     Figures are of low quality. The information in Figure 2 is not clearly expressed. "Figure 7(e) Test MGP1, vitamin E " changed to "Figure 7 (f) Test MGP1, vitamin E". And the article format needs further checking. The format of the numbers in the table should be uniform.

Reviewer 2 Report

In this manuscript, hydrodynamic cavitation processing of almond is investigated. Introduction is written with relevant and interesting data, providing justification for study, as well as the importance of investigated subject. Manuscript represents a basis for further studies. Discussion is written clearly with relevant literature included.

There are some minor suggestions:

„However, it is anticipated that the recovery and quantification of polyphenols from raw almond materials during laboratory analysis is critically dependent on the details of the method and could be affected by greater systematic uncertainties than declared [31], an issue that does not affect the analysis of clear aqueous extracts.“ Does this mean the results are not reliable? Please clarify better.

Instead of “level”, which is confusing in the context of compounds, content or something similar will be more suitable.

“Energy level: EU Regulation No. 1169/2011 of the European Parliament and of the Council of 25 October 2011 on the provision of food information to consumers (https://eur-lex.europa.eu/eli/reg/2011/1169/ojv);“ Page doesn’t exist.

Additionally, the majority of the pages written in the section Microbiological, nutritional and vitamin analyses should be checked. If you cannot provide the adequate page, please include the entire method of analysis. Methods must be written in a way that allows replication.  

It is not described in Methods how Energy Consumption was measured.
